# Microstructural Investigation of Process Parameters Dedicated to Laser Powder Bed Fusion of AlSi7Mg0.6 Alloy

**DOI:** 10.3390/ma17092156

**Published:** 2024-05-05

**Authors:** Janusz Kluczyński, Tomáš Dražan, Zdeněk Joska, Jakub Łuszczek, Robert Kosturek, Katarzyna Jasik

**Affiliations:** 1Institute of Robots & Machine Design, Faculty of Mechanical Engineering, Military University of Technology, Gen. S. Kaliskiego St., 00-908 Warsaw, Poland; jakub.luszczek@wat.edu.pl (J.Ł.); robert.kosturek@wat.edu.pl (R.K.); katarzyna.jasik@wat.edu.pl (K.J.); 2Department of Mechanical Engineering, Faculty of Military Technology, University of Defence, 662 10 Brno, Czech Republic; tomas.drazan@unob.cz (T.D.); zdenek.joska@unob.cz (Z.J.)

**Keywords:** additive manufacturing, metallography, porosity, microhardness, AlSi7Mg0.6

## Abstract

This study presents a microstructural investigation of the printing parameters of an AlSi7Mg0.6 alloy produced by powder bed fusion (PBF) using laser beam melting (LB/M) technology. The investigation focused on the effects of laser power, exposure velocity, and hatching distance on the microhardness, porosity, and microstructure of the produced alloy. The microstructure was characterized in the plane of printing on a confocal microscope. The results showed that the printing parameters significantly affected the microstructure, whereas the energy density had a major effect. Decreasing the laser power and decreasing the hatching distance resulted in increased porosity and the increased participation of non-melted particles. A mathematical model was created to determine the porosity of a 3D-printed material based on three printing parameters. Microhardness was not affected by the printing parameters. The statistical model created based on the porosity investigation allowed for the illustration of the technological window and showed certain ranges of parameter values at which the porosity of the produced samples was at a possible low level.

## 1. Introduction

Additive manufacturing (AM) has become an increasingly popular method that continues to transform the manufacturing industry [1,2,3]. Today’s additive technologies have advanced capabilities regarding material options, speed, precision, cost-effectiveness, and the possibility of producing complex geometries with high accuracy [4]. AM shows promise for the production of components with alloys that cannot be processed with traditional processes and also unique microstructure [5,6]. Each of the dozens of additive technologies has a large number of possibilities but also has limitations, and therefore, it is necessary to select the appropriate additive technology to obtain the desired product properties [7]. Some of the common problems faced by most additive technologies are porosity, hot cracking, the anisotropy of the mechanical properties, and surface quality. The investigation of mechanical properties and microstructural investigation is essential to optimize the printing parameters and achieve high-quality components [2,6,7,8,9,10,11,12,13]. Powder bed fusion-selective laser melting (PBF-LB/M) is one of the most widely used additive manufacturing technologies, where metal powder is melted layer-by-layer using a laser to create the desired component [2,9,10,11,14,15,16]. In recent years, there has been a growing interest in the use of PBF-LB/M to produce components for aerospace, automotive, biomedical applications, and a lot of other branches [4,5,12,16,17]. PBF produces a specific microstructure by melting the material and making a melting pool in the printing direction. Heat directed to the melting point is distributed through the build material by thermal conduction to form an oriented fine-grained microstructure [7,17]. Therefore, microhardness measurements are often used to correlate the microstructure with the mechanical properties of the printed components [14]. The parts produced by SEM typically require post-processing to achieve the desired surface finish and mechanical properties [18]. Post-processing can include removing support structures, sandblasting or polishing, and heat treatment to reduce residual stress [14]. Of course, the post-processing steps are tailored to the specific material and application requirements. Aluminum alloys, such as AlSi7Mg0.6, are widely used materials in these industries due to their strength-to-weight ratio and corrosion resistance [14]. However, the quality of the final product is highly dependent not only on powder quality [19] and heat treatment but especially on printing parameters, including laser power, hatching distance, and exposure velocity [7,17]. Optimizing these printing parameters for this material is challenging because each parameter affects a lot of variables [1,13]. The idea of creating a mathematical model showing the theoretical porosity using these printing parameters seems to be very advantageous. Aluminum materials are known for their high thermal conductivity and, thus, the occurrence of residual stress and warping [1,9,20]. Today’s aluminum alloy research deals with printed PBF-LB/M technologies, mostly addressing quality depending on the printing orientation [4,8,11], research on material properties and production of metal powder [3,21], the formation of defects, corrosion, stress corrosion, and the effect of heat treatment [4,5,10,18,20,22] or by modeling and predicting the properties of printed parts [17]. In the case of research on the effect of changing the printing parameters, the articles bend in the range of different values of printing parameters than in our research [1,6] or dealing with other specifications [12,13,15,16,22,23]. Martucci et al. [1] conducted research on the printing strategies of the AlSi10Cu8Mg aluminum alloy using laser powers below 100 W and lower printing speeds. His results indicated an improvement in terms of porosity but difficulties with the problem of crack formation and delamination. By optimizing the process parameters, he was able to produce high-quality samples. Arvieu et al. [6] then dealt with the interpretation of the relative density of AlSi7Mg0.6 and AM205 aluminum alloys printed by PBF-LB/M.

The aluminum alloy sample investigated in our work consists of only five different printing processes with significantly different parameters. However, the results of his research show that the energy density has a direct influence on the porosity of the printed sample, and it is possible to achieve relative densities even higher than the 100% reported in the literature. No research has been found that has attempted to develop a mathematical model of the printing parameters and their effect on the porosity of the AlSi7Mg0.6 alloy produced by PBF-LB/M technology. In the present state of the art, there is a small amount of research related to microstructural evaluation with this range of processing parameters of PBF-LB/M technology and implementing them to mathematical models. Therefore, understanding energy density combined with the dependent printing parameters and their influence on the porosity and microstructure of the printed components is crucial for designing and producing components with the best possible material and mechanical properties.

The aim of our research is to develop a comprehensive understanding of the relationship between energy density and various printing parameters in the production of aluminum alloy components using PBF-LB/M technology. Specifically, we aim to investigate how different combinations of printing parameters affect the porosity and microstructure of AlSi7Mg0.6 alloy samples. By analyzing the influence of energy density along with other printing parameters, we seek to develop a mathematical model that accurately predicts the porosity levels and microstructural characteristics of printed components. This research is essential for optimizing the manufacturing process and achieving components with superior material and mechanical properties.

## 2. Materials and Methods

The 10 mm cube-shaped samples were designed in Magic 19 software (Leuven, Belgium, Materialise), and the same software was used to provide printing parameters, such as laser power, layer thickness, exposure velocity, and hatching distance (clearance between laser irradiation lines). The number of geometries was added to the top of the cube for easy recognition of the samples and to avoid confusion between the samples. These cubes were spaced around the print bed so that they would not interfere with each other during the process, as shown in Figure 1. The printing of the samples was performed on an SLM 125HL 3D PBF-LB/M printer (SLM Solution, Lubeck, Germany). Before the process, the equipment was properly cleaned of previous processes and materials. Before loading and calibration, the material was dried in a SUSLAB-BIO-005 laboratory dryer (Adverti, Łódź, Poland) for 24 h at 90 °C temperature. An aluminum alloy AlSi7Mg0.6 (AC-42200), gas atomized powder supplied by SLM Solutions (SLM Solution, Lubeck, Germany) was used as the base material. The powder particles were characterized by a spherical particle size of 20–63 µm. The thermal conductivity of this material was 150–170 W/(m·K). The chemical composition of this material, according to the material data sheet, is given in Table 1 below. The quality of the metal powder was checked by a scanning electron microscope (SEM) Jeol JSM-6610 (Jeol, Tokyo, Japan).

The observed process parameters selected were laser power, exposure velocity, and the hatching distance. The thickness of the powder layer was fixed with a constant value of 0.03 mm. The parameter which is a function of these 3 parameters is the energy density. For this, Equation (1) was used.
(1)$$E_{V}=\frac{P_{L}}{h\times v_{s}\times H}$$

E_V_—energy per unit volume (J/mm^3^);

P_L_—laser power (W);

h—hatching distance (mm);

v_s_—exposure velocity (mm/s);

H—thickness of the powder layer (mm).

Many combinations of these parameters were created for the mathematical model and are published in Table 2. In total, 37 test samples were printed. Three groups of samples were created according to the dominant parameters. The first set (1–13) were samples printed with high laser power (>300W) and high exposure velocity (>1300 mm/s), while the second set (14–25) were printed with laser power (<200W) and hatching distance (<0.1), and the third set (26–37) with laser power (<200W) and hatching distance (<0.1).

To avoid thermal influence during the removal of the samples from the printing substrate, wire electric discharge cutting technology was used on the Accutex AL 400SA machine (AccuteX Technologies Co., Ltd., Taichung City, Taiwan). For experiments and evaluation, samples were cut in the cross-section through the whole layers of each sample by means of a metallographic saw with direct cooling. Subsequently, the samples were mounted in resin and subjected to grounding and polishing. Porosity measurements were performed on all 37 polished samples using a Keyence (Osaka, Japan) VHX-7000 digital microscope. An analysis of the structural quality of the samples produced using different printing parameters was divided into two stages. In the first step, there was stitching of sequentially exposed areas, and then the ratio of the porous area detected through the optical device to the total area of the sample was determined. The measurements for all specimens were performed using the same settings related to the detection of defects. The porosity value presented in this manuscript is based on a single measurement for each plane. Afterward, the specimens were re-polished and etched for metallographic evaluation. Mounted and polished samples were etched with Keller’s reagent used for aluminum alloys. It is a mixture of nitric acid (HNO_3_), hydrochloric acid (HCl), and hydrofluoric acid (HF). The etching times were significantly different depending on the printing parameters, and thus, multiple sample preparations were necessary. Multiple images were taken of each sample at three magnifications for a complex evaluation. The final test was the measurement of microhardness on an automated Struers DuraScan hardness tester (Struers, Copenhagen, Denmark) using the Vickers method. 

Due to the material properties, the measurements were made under a load of 100 g and evaluated with a microscope at 40x magnification. The measurements were made with 6 indentations at 1 mm from the edge and 4 from the individual indentations. Before each indent, the position was checked and corrected to avoid local porosity. 

Due to the time-consuming nature of the parameter selection process, a decision was made to attempt the use of a Design of Experiment (DoE) analysis. The mathematical models from the DoE area, with their predictive capabilities for the dependent variable, have the potential to reduce the time required for parameter selection by minimizing the number of parameter groups that need to be tested. Consequently, it decreases the time intensity of such research endeavors. The authors of [25,26] adopted such an approach in the parameter selection process for manufacturing model elements using the PBF-LB/M technique. One of the types of models in this context is the quadratic surface regression model. This model amalgamates polynomial regression models with fractional models. It takes into account not only the influence of independent variables on the predictor but also the interactions among variables. The independent variables encompass the components of energy density derived from formula (1), except for the layer thickness, which remained a constant value. Meanwhile, the described value (dependent variable/predictor) is the porosity value. The general form of the model is presented in formula (2):(2)$$y=\beta_{0}+\beta_{1}x_{1}+\beta_{2}x_{2}+\beta_{3}x_{3}+\beta_{11}x_{1}^{2}+\beta_{22}x_{2}^{2}+\beta_{33}x_{3}^{2}+\beta_{12}x_{1}x_{2}+\beta_{13}x_{1}x_{3}+\beta_{23}x_{2}x_{3}+\in$$

In Equation (2), the individual components represent the following:

y—porosity;

x_1_—laser power;

x_2_—exposure velocity;

x_3_—hatching distance;

β_m_ and β_mn_ (for m = 1,2,3; n = 1,2,3)—the regression coefficients for individual variables and their product combinations;

∈—modeling residual error.

The source data for the model comprises actual porosity measurements from samples produced using the groups listed in Table 2. However, for a better fit of the model to the source values (as described by the coefficient R^2^), a portion of parameter groups was excluded, where the values of a given independent variable did not repeat in at least three other groups. The excluded groups were as follows: 2, 4–8, 10–13, 26–28, 31–32. The necessity of employing this approach arises from a significant increase in the model prediction error (∈) due to the utilization of individual parameter groups where the values of specific independent variables do not recur in other groups. Additionally, a *p*-value was determined. The estimation of the p coefficient involved created an analysis of variance (ANOVA) table, and its relevance pertained to conducting statistical tests. Each component’s statistical significance was assessed using a threshold of *p* < 0.05. All calculations and charts were made using the Statistica software (TIBICO Software Inc., Palo Alto, CA, USA). The purpose of using this model is to determine the group of parameters according to which the porosity value of the model elements is close to 0%. The results of the statistical model serve as a source of groups of parameters that can be tested in further stages of work on the discussed material. 

## 3. Results

Particle analysis of the powder used for printing the samples was performed on a Tescan Mira4 scanning electron microscope (Tescan Brno Czechia). Figure 2 shows a relatively uniform powder particle size distribution at lower resolution. The size of the particles varied from 2 to 30 µm, and they had a spherical or slightly ellipsoidal shape. In the detailed image (Figure 2—right side), a spherical particle with a diameter of 40 µm is shown with fused small satellites with a diameter of approximately 3 µm, which indicates that recycled powder was used for printing, where both new and used powders were used. 

### 3.1. Porosity Measurement

A crucial factor in Metal Additive Manufacturing (MAM) methods that allow the quality verification of produced parts is porosity [26,27,28,29,30,31], which is why this parameter was used for the first step selection of the produced parts for further analysis. The percentage of porosity in all produced samples ranged from 0.38% to 1.71%. On average, the value of pore representation was 1%, with a standard deviation equal to 0.29%. The area of the measured pores was very small, and therefore, the observation of maximum and average perimeter sizes from all pores was chosen. The maximum measured circumference was 3.71 mm, and the smallest was 0.31 mm. The average perimeter for the samples was 0.1 mm, and the smallest average perimeter was 0.04 mm. Thus, the majority of pores had a perimeter limit close to 0.31 mm. Sample No. 22 was the one with the highest proportion of porosity, equal to 1.36% (shown in Figure 3).

A properly made drying process made the gas porosity (caused by moisture) marginal. Visible increased porosity on the side parts of the sample was caused by the same outline perimeter shell parameters used for all samples. For the measurement, these porosities were not taken into account. Only green-highlighted pores were considered in porosity calculations. As is visible in Figure 3, irregular voids are mostly caused by a lack of fusion phenomenon [32,33]. This factor is strictly dependent on process parameters, which is why the number of this kind of void fluctuation depends on the process parameters used. 

Due to the large volume of samples created and the large number of possible combinations of the influence of individual printing parameters, two groups of samples (Table 3) were selected for this article, which differed significantly in certain process parameters but were the same or similar in importance for others. The first group of samples that were selected for metallographic evaluation were samples 12 and 36, which had the lowest and highest energy density of the entire group of samples.

In these samples, the laser power differed by only 20 W, with the main difference appearing in the different exposure velocities, which was one-third higher in the sample with lower energy density, and the hatching distance, which was almost twice as high in the sample with lower energy density. The microstructure of the No. 25 and No. 36 samples is shown in Figure 4 and Figure 5.

Both samples show a similar microstructure, which is typical for 3D printing with bands of melted pools of irregular shape. Despite some similarities at the macrostructural level, the sample produced at a higher energy density was characterized by a clearly smaller size of melted pools. The established size of the melted pools equaled 32.4 ± 7.3 μm and 23.1 ± 7.1 μm for the samples with a lower and higher energy density, respectively. These results indicate a reduction in the average size of about 30%. On the microstructural level, the lower energy density is reflected in the slightly finer columnar cellular substructure and the presence of ultrafine equiaxed columns with a width of about 8 μm and cell size below 1 μm. The increase in energy density causes the columnar substructure to coarse, and at the same time, no ultrafine equiaxed cells in the zone were found. In addition, in the sample with a higher energy density, the presence of microcracks oriented perpendicularly to the direction of columnar grain growth was reported, which suggests their solidification nature.

### 3.2. Metallography—Same Energy Density

In the case of using the same energy density, samples number 2 and 18 (shown in Table 4) were chosen for further analysis, which have an energy density of 50 J/mm^3^. Their selection was justified to analyze how different laser powers affect the microstructure. Such a comparison is very important from an economic point of view. 

The etching of microstructure (Figure 6 and Figure 7) by Keller’s reagent for samples produced by laser power at around 350 W took twice the time compared to samples produced under laser power at around 180 W. Sample 2 came out on top with the power of 350 W while sample 18 exhibited a power of 180 W in magnification. 

Comparing the samples obtained with various levels of laser power and constant energy density, the conclusion can be drawn that their macrostructures are very similar in terms of melted pool sizes. Despite the presence of porosity, observations did not reveal a noticeable number of other imperfections (e.g., microcracks). The microstructures of the investigated samples exhibited similar features, with the width of melted pool interfaces in both cases at about 10 μm wide. The only visible difference was the grainy structure of the interfaces and the fact that the sample with lower power (180 W) was characterized by finer grains in this area. The participation of equiaxed grains at the expense of columnar grains was also higher than for the sample obtained with 350 W. Referring these observations to the results for the samples obtained with different values of energy density, the conclusion can be drawn that the value of energy density had the highest impact on the macrostructural level (melted pool size, imperfections, melted pool interface width). At the same time, with an optimized energy density, the laser power can shape the microstructure of the AlSiMg alloy. Based on the performed observations, it can be stated that a lower value of laser power (180 W) is more profitable in terms of grain size in the melted pool interface for investigated energy density (about 50 J/mm^3^).

The image from sample number 36 (Figure 8) shows the association of incompletely melted powder with a coarse-grained structure and columnar grains oriented in accordance with the heat flow direction. In the image of sample 29 (Figure 9), the boundary between the fine-grained and coarse-grained structures was separated by a crack. The columnar grains, oriented in accordance with the direction of heat flow, crystallized from the probable point of contact of the two interfaces.

In sample number 26 (Figure 10), two solidification cracks separated the fine-grained zone from the area of the columnar grain, whose growth was dictated by a small, 10 μm width bridge allowing for further heat flow. The image of sample 34 (Figure 11) then indicates that porosity formed at the boundary of the melting pools.

In sample 3 (Figure 12 and Figure 13), microcracks initiated from a spherical pore formed by moisture in the matrix were observed. Similarly, a crack was observed in the middle of the melting pool. These are probably the thermal cracks that are commonly found at the boundaries of melting pools. 

### 3.3. Microhardness Measurement

Hardness measurement results are presented on the chart in Figure 14. The average hardness value was 128 HV 0.1 with a standard deviation of 2.5 HV 0.1. The majority of results for individual parameter/sample groups fell within this range of deviation. Hence, it can be concluded that within the investigated range of parameter values, no significant changes in properties in relation to material hardness occurred. These findings are further supported by microstructural analysis results. The dotted red line indicates the mean value for all measurements.

In reference to the literature, other researchers achieved hardness levels ranging from 45 to 120 HV, with power density values ranging between 10 and 242 J/mm^3^ [14,34]. These hardness values were lower by approximately 8 HV compared to the average value obtained in this study and the maximum value found in the literature. However, the direct comparison of these values is challenging as numerous other additional factors, such as exposure strategy or layer height, which were not considered in this study, could significantly influence the final outcome. The settings utilized were default settings provided by the machine manufacturer. Moreover, based on this study and existing data from the literature, it can be inferred that without additional post-processing, or other supplementary procedures related to the process (such as the re-exposure of layers), achieving values significantly higher than approximately 120–130 HV is not feasible. Furthermore, when comparing the values of additively manufactured components with conventionally manufactured components from the same material, the AlSi7Mg0.6 aluminum alloy conventionally manufactured and achieved hardness levels above 120 HV only after additional heat treatment, precisely in the T6 state [35]. Hence, in the case of components manufactured using PBF-LB/M from aluminum alloys, it is feasible to eliminate certain processes associated with heat treatment if the demonstrated level of hardness is desirable.

### 3.4. Statistical Model

Based on the analysis of 22 parameter groups, analytical calculations were conducted to determine the regression coefficients β_m_ and β_mn_ (for m = 1,2,3; n = 1,2,3), along with the determination of the parameter value p. This facilitated the derivation of the final form of the mathematical model (quadratic surface regression), as represented by Equation (3). Furthermore, the coefficient of correlation R^2^ was determined to be 0.85.
(3)$$y=6.9490-0.0735x_{1}+0.0078x_{2}-23.9161x_{3}+0.0003x_{1}^{2}+0.000003x_{2}^{2}+15.9355x_{3}^{2}-0.0001x_{1}x_{2}+0.0827x_{1}x_{3}-0.0088x_{2}x_{3}+0.4390$$

In Table 5, the measured porosity values for the modeled elements produced using 37 parameter groups are presented. Additionally, the model, which indicated values for the 22 considered parameter groups in the DoE analysis, was also juxtaposed. The rest of the parameter groups (2, 4–8, 10–13, 26–28, 31–32) were taken for the validation of the created statistical model. The largest disparity between the mathematical model and the actual value occurred in group 35, with a difference of 0.36%. The mathematical model identified group 14 (out of 22 considered) as the parameter set, resulting in the lowest porosity for the modeled elements, aligning with actual measurements. An important consideration is that porosity measurement results for samples manufactured using the same energy density from different parameter groups exhibited variation. This clearly signals that when specifying manufacturing parameters for particular materials, it is crucial to provide not only energy density and layer thickness but also other parameters, such as exposure velocity, laser power, and the distance between hatching lines.

The mathematical model also enabled the generation of response surfaces (Figure 15), assuming that one value of the independent variable remained constant. The selected constant value was the distance between irradiation vectors, as this is the parameter least frequently subject to adjustments among the three considered. Below are presented five response surfaces for five values of the distance between irradiation vectors, which were among the 22 groups under consideration. 

The remaining values of the two variables on the *X*-axis (laser power) and *Y*-axis (exposure velocity) were constrained within the predetermined upper limits. Specifically, these were 400 W due to the maximum achievable power on the machine used to produce the samples and 2000 mm/s, as higher values proved impractical in the process due to the phenomenon of ‘balling’, which may occur at such high speeds.

The orientations of fields indicating the lowest porosity values (<1%) exhibited a similar configuration in each case. They fell within the range of exposure velocities from 1200 mm/s to 2000 mm/s, while the laser power values ranged approximately from 225 W to 375 W. Each surface response distinctly delineated a technological window enabling the fabrication of elements without process failure risks. The position of this technological window, contingent upon the distance of values between irradiation vectors, is inclined at various angles to the *X*-axis. 

This inclination increases with higher distance values between irradiation vectors. As the distance between laser paths increases, the power density decreases, consequently shifting the window model towards higher laser power values. It is crucial to note that the model’s extrapolation of values beyond the considered parameter range in the 22 groups introduces significant error. Moreover, the theoretical minimum of function (4) represents a group of parameters, ostensibly allowing for the fabrication of elements with a porosity close to 0%. This group is detailed in Table 6.

The power density achieved from the proposed set of parameters amounted to 43.95 J/mm^3^. Samples produced at similar power density values exhibited porosity levels of 0.79% and 1.16%. However, these were groups characterized by distinct values of individual components. Due to the limited scope of this article, this particular group will be further examined in subsequent research endeavors related to its utilization.

## 4. Conclusions

A properly adjusted process is the key to efficient and economical production and printing success, but above all, the quality of the microstructure and optimal mechanical properties of the printed parts. A wide range of process parameters of the PBF-LB/M technology for aluminum alloy AlSi7Mg0.6 were compared, and a mathematical model was developed to simplify the search for optimal parameters. Based on the obtained results, the following conclusions were drawn:Using high laser power (>300 W), there were no unmelted particles of material; on the other hand, thermal cracks were found.When using low laser power settings (<200 W) and high energy densities, paradoxically, a large number of large pores formed, and incorrectly melted powder particles also formed.Microhardness was not affected by the change in process parameters and reached a very narrow range of values.The quadratic surface regression model perfectly illustrates the technological windows and shows certain ranges of parameter values at which the porosity of the produced samples was low.Within the considered range of 22 groups of parameters, regarding the group of manufacturing parameters for which the samples showed the lowest porosity, the model indicated the same group, No. 14, which was characterized by the smallest number of defects in the case of actual measurements.

## Figures and Tables

**Figure 1 materials-17-02156-f001:**
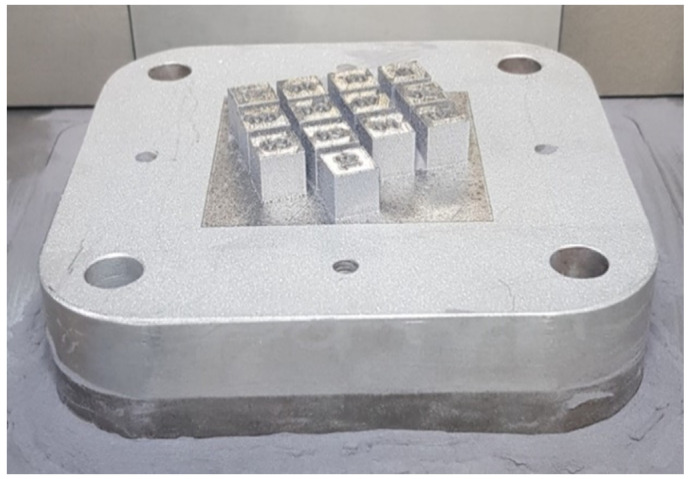
Distribution of printed samples on the substrate plate.

**Figure 2 materials-17-02156-f002:**
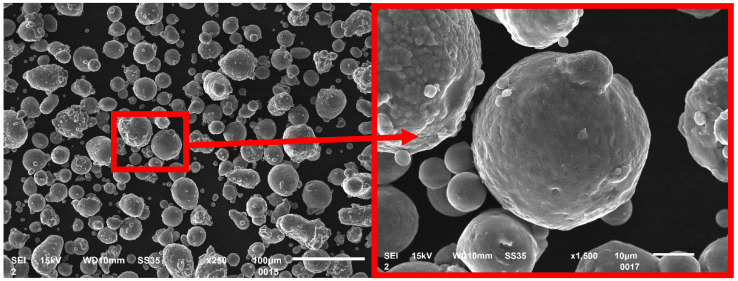
SEM image of the powder particles used to print the samples.

**Figure 3 materials-17-02156-f003:**
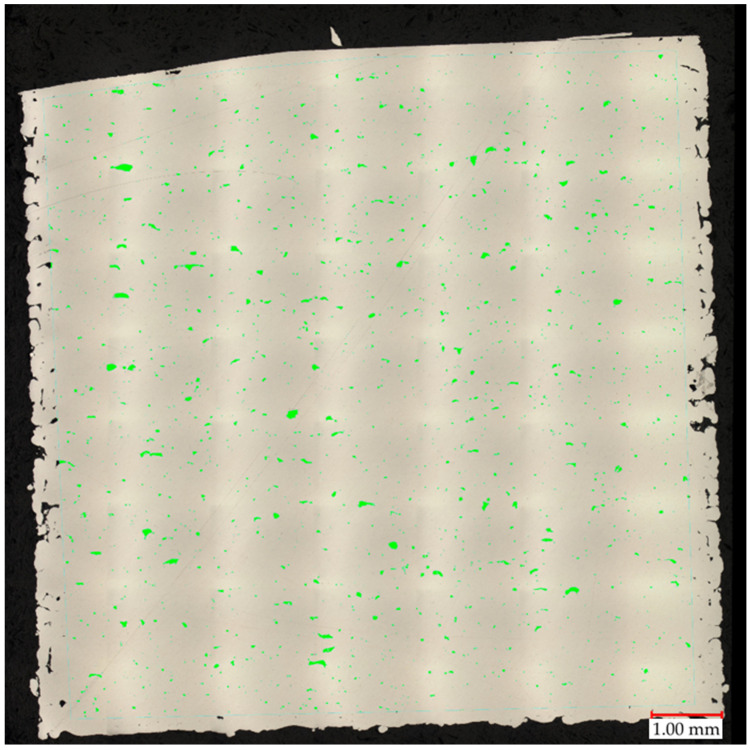
Merged images on areal porosity of sample 22 characterized by the highest porosity.

**Figure 4 materials-17-02156-f004:**
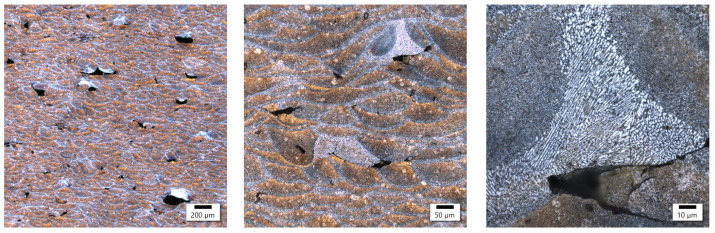
Sample 25’s microstructure at different magnifications.

**Figure 5 materials-17-02156-f005:**
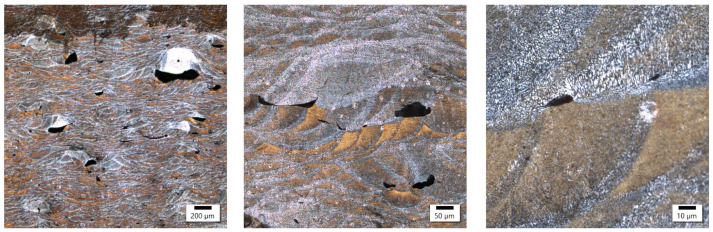
Sample 36’s microstructure at different magnifications.

**Figure 6 materials-17-02156-f006:**
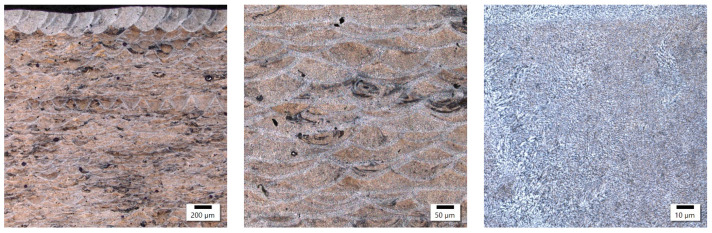
Sample 2’s microstructure at different magnifications.

**Figure 7 materials-17-02156-f007:**
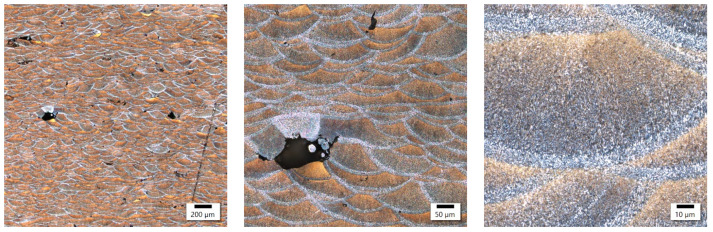
Sample 18’s microstructure at different magnifications.

**Figure 8 materials-17-02156-f008:**
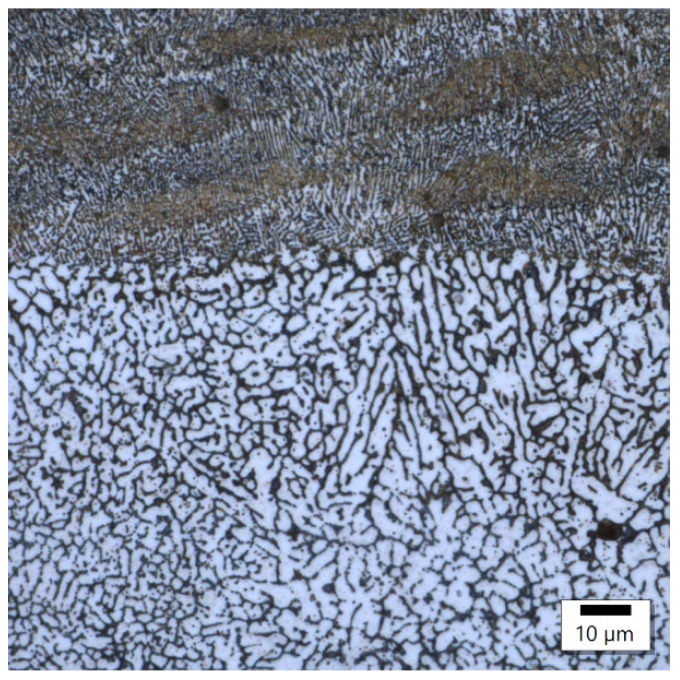
Sample 36’s coarse-grained structure.

**Figure 9 materials-17-02156-f009:**
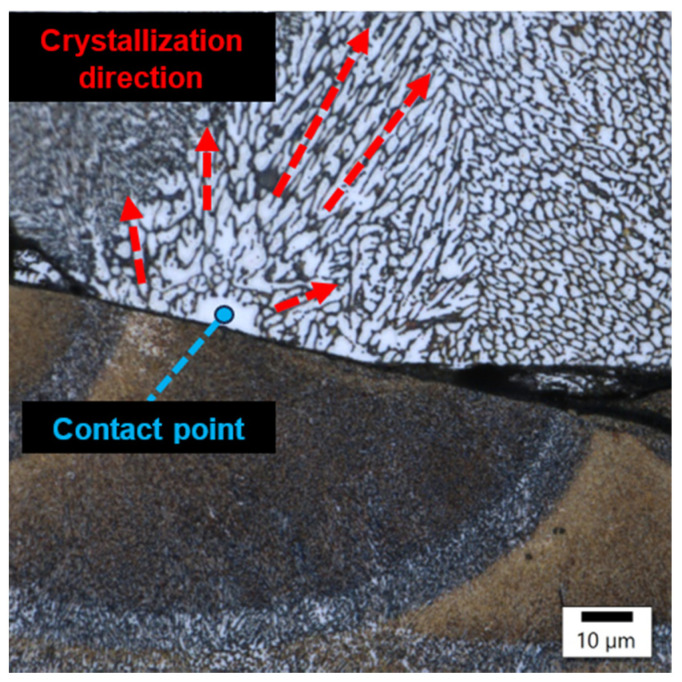
Sample 29’s point of contact of imperfectly melted material.

**Figure 10 materials-17-02156-f010:**
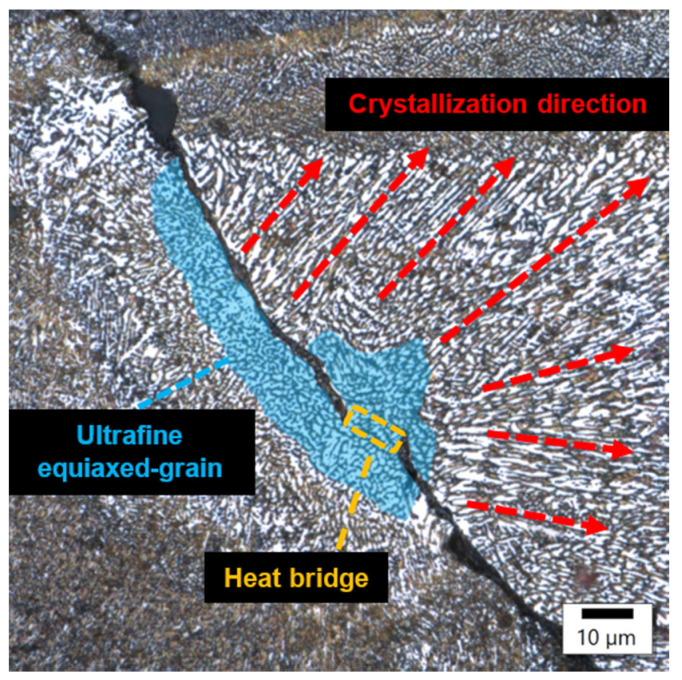
Sample 26 showing cracks in the structure due to heat bridge.

**Figure 11 materials-17-02156-f011:**
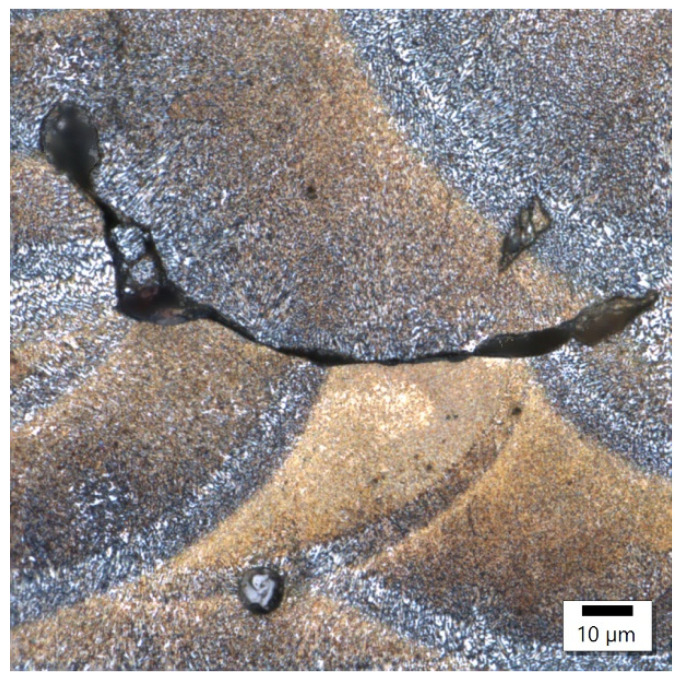
Sample 34 showing porosity on the boundaries of melting pools.

**Figure 12 materials-17-02156-f012:**
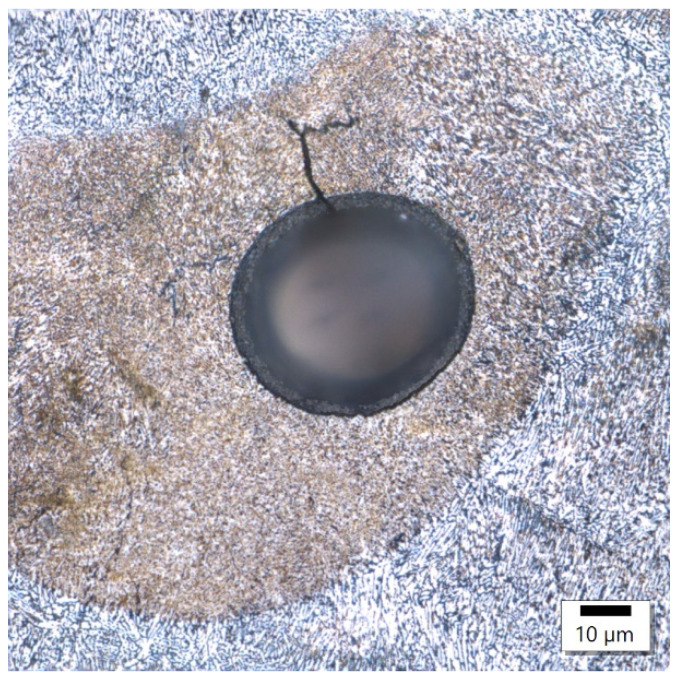
3 Sample 3’s spherical pore.

**Figure 13 materials-17-02156-f013:**
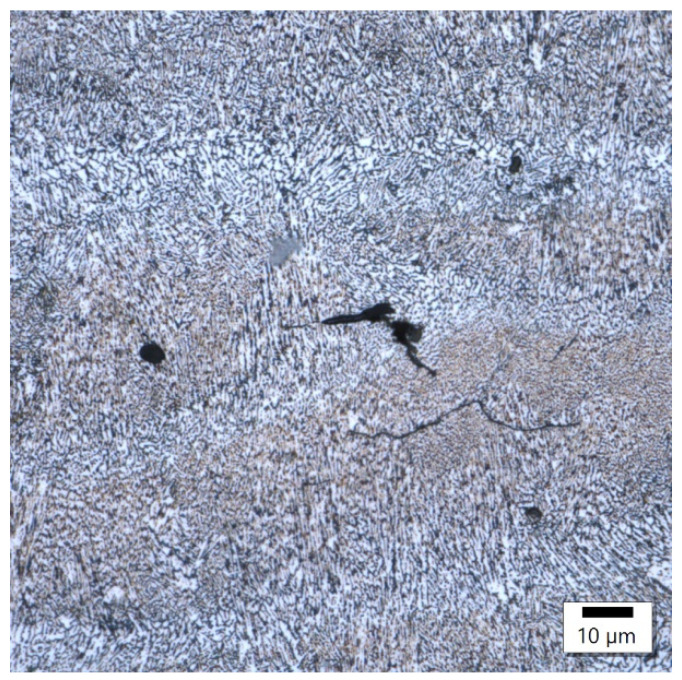
Sample 3 showing cracks in the middle melting pool.

**Figure 14 materials-17-02156-f014:**
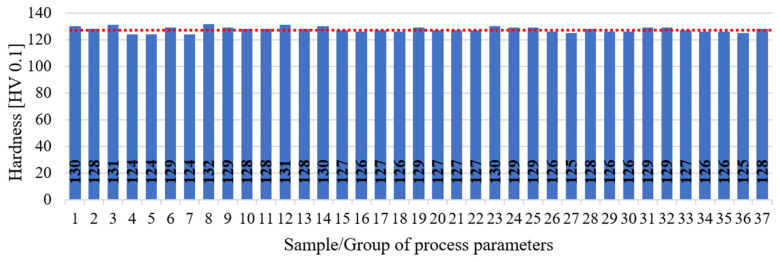
The results of measurement tests of AlSi7Mg0.6 samples produced by means of PBF-LB/M.

**Figure 15 materials-17-02156-f015:**
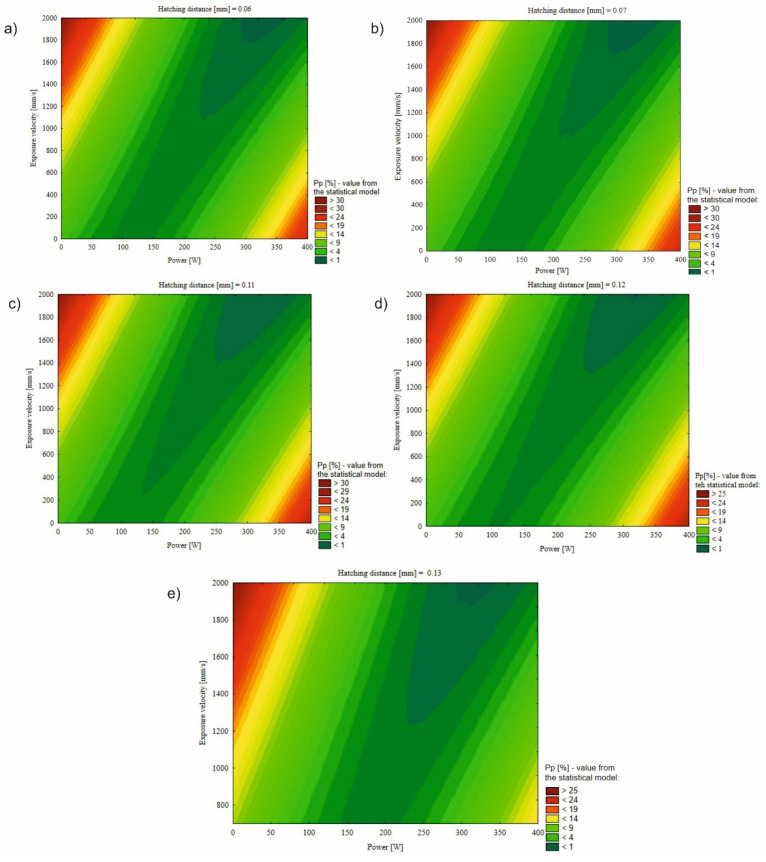
Surface response maps of the quadratic regression model for the constant hatching distance: (**a**) 0.06, (**b**) 0.07, (**c**) 0.11, (**d**) 0.12, and (**e**) 0.13.

**Table 1 materials-17-02156-t001:** Chemical composition of AlSi7Mg0.6 in wt.% [24].

Al	Cu	Fe	Mg	Nb+Ta	Mn	Si	Ti	N	Zn	Other Each	Another Total
Bal.	0.05	0.19	0.45–0.70	/	0.1	6.50–7.50	0.25	/	0.07	0.03	0.10

**Table 2 materials-17-02156-t002:** Combination of printing parameters.

Sample	Power P [W]	Layer Thickness [mm]	Exposure Velocity [mm/s]	Hatching Distance [mm]	Energy Density [J/mm^3^]
1	350	0.03	1650	0.13	54.390
2	350	0.03	1650	0.14	50.505
3	350	0.03	1650	0.12	58.923
4	385	0.03	1650	0.13	59.829
5	315	0.03	1650	0.13	48.951
6	350	0.03	1815	0.13	49.446
7	350	0.03	1485	0.13	60.433
8	350	0.03	1650	0.15	47.138
9	350	0.03	1650	0.11	64.279
10	400	0.03	1650	0.13	62.160
11	280	0.03	1650	0.13	43.512
12	350	0.03	1980	0.13	45.325
13	350	0.03	1320	0.13	67.988
14	190	0.03	800	0.13	60.897
15	180	0.03	800	0.13	57.692
16	170	0.03	800	0.13	54.487
17	190	0.03	1000	0.12	52.778
18	180	0.03	1000	0.12	50.000
19	170	0.03	1000	0.12	47.222
20	190	0.03	1000	0.11	57.576
21	180	0.03	1000	0.11	54.545
22	170	0.03	1000	0.11	51.515
23	190	0.03	1200	0.11	47.980
24	180	0.03	1200	0.11	45.455
25	170	0.03	1200	0.11	42.929
26	190	0.03	1200	0.1	52.778
27	190	0.03	1200	0.09	58.642
28	190	0.03	1200	0.08	65.972
29	190	0.03	1200	0.07	75.397
30	190	0.03	1200	0.06	87.963
31	190	0.03	1200	0.05	105.556
32	190	0.03	1000	0.05	126.667
33	190	0.03	1000	0.06	105.556
34	190	0.03	1000	0.07	90.476
35	190	0.03	800	0.07	113.095
36	190	0.03	800	0.06	131.944
37	190	0.03	800	0.07	113.095

**Table 3 materials-17-02156-t003:** Process parameters used for further microstructural analysis.

Sample	Power [W]	Layer Thickness [mm]	Exposure Velocity [mm/s]	Hatching Distance [mm]	Energy Density [J/mm^3^]	Measured Porosity (%)
25	170	0.03	1200	0.11	42.929	1.67
36	190	0.03	800	0.06	131.944	1.53

**Table 4 materials-17-02156-t004:** Samples 2 and 18 and their printing parameters.

Sample	Power [W]	Layer Thickness [mm]	Exposure Velocity [mm/s]	Hatching Distance [mm]	Energy Density [J/mm^3^]	Measured Porosity (%)
2	350	0.03	1650	0.14	50.505	0.78
18	180	0.03	1000	0.12	50.000	1.05

**Table 5 materials-17-02156-t005:** Measurements of porosity. Maximum and average perimeters and porosity predicted by the statistical model.

Sample	Measured Porosity (%)	(Pρ) [%]—Statistic Model Equation (3)	Measured Max Perimeter [mm]	Measured Average Perimeter [mm]
1	0.75	0.72	0.33	0.05
2	0.78	-	0.35	0.05
3	0.70	0.77	0.35	0.05
4	0.75	-	0.36	0.06
5	0.90	-	0.48	0.06
6	1.01	-	0.42	0.05
7	0.92	-	0.37	0.06
8	0.71	-	0.31	0.05
9	0.87	0.83	0.44	0.06
10	1.04	-	0.43	0.06
11	0.79	-	0.45	0.06
12	1.16	-	0.38	0.05
13	1.45	-	0.53	0.06
14	0.38	0.43	0.63	0.04
15	0.48	0.47	0.39	0.05
16	0.61	0.57	1.41	0.06
17	0.46	0.53	0.84	0.05
18	1.05	0.70	0.97	0.06
19	0.64	0.93	0.59	0.06
20	0.77	0.66	0.64	0.06
21	0.86	0.85	1.47	0.06
22	1.09	1.09	1.00	0.06
23	0.74	0.90	0.56	0.06
24	1.21	1.22	3.71	0.06
25	1.67	1.59	0.77	0.06
26	1.34	-	0.93	0.06
27	1.06	-	0.82	0.06
28	0.98	-	0.56	0.06
29	1.71	1.54	1.29	0.08
30	1.62	1.71	1.62	0.09
31	1.32	-	1.04	0.07
32	1.42	-	1.70	0.09
33	1.25	1.38	1.17	0.08
34	1.27	1.23	0.85	0.08
35	0.80	1.16	0.57	0.07
36	1.53	1.29	2.02	0.10
37	1.28	1.16	1.35	0.08

**Table 6 materials-17-02156-t006:** Predicted group of parameters using a function (4)—minimum of the function.

Laser Power [W]	Exposure Velocity [mm/s]	Hatching Distance [mm]	Layer Height [mm]	Predicted Porosity [%]
282.86	1650	0.13	0.03	≈0%

## Data Availability

The research data are available upon request (via janusz.kluczynski@wat.edu.pl).

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
