# Peer review of "Microstructural Investigation of Process Parameters Dedicated to Laser Powder Bed Fusion of AlSi7Mg0.6 Alloy"

_materials, 2024, doi:10.3390/ma17092156_

Round 1

Reviewer 1 Report

Comments and Suggestions for Authors

The reviewer appreciates the effort of the author on this manuscript. This study is well designed, conducted and executed well. The manuscript is well written. However, after careful reading, the reviewer has several comments and recommendations.

Title-  please use the full form of PBF-LB/M AlSi7Mg0.6

Abstract- The conclusion “The results of this study can be used to optimize the print-20 ing parameters for producing high-quality AlSi7Mg0.6 components using PBF-LB/M technology.” Needs to be revised. The conclusion should be the specific outcome of the study rather than a general recommendation.

Methodology- Add a description of “The 10 mm cube-shaped test bodies and printing conditions were created in Magic 85 19 software (Leuven, Belgium, Materialise).”

Please revise this paragraph “For the experiment was used aluminum alloy AlSi7Mg0.6 (AC - 42200) was from SLM Solutions with a spherical particle size of 20 - 63 μm. The thermal conductivity of this material is 150 - 170 W/(m·K). The chemical compo- sition of this material according to the material data sheet is given in Table 1 below. The quality of the metal powder was checked by scanning electron microscope (SEM) Jeol JSM-6610 (Jeol, Tokyo, Japan).

The author has listed 37 printing parameters in Table 2. Does it mean the number of test groups is 37? How many specimens were tested in each group (n=?)

The reviewer’s opinion the number of specimens in each test condition should be a minimum of 3 to enable statistical comparison and to draw a specific conclusion.   

Comments on the Quality of English Language

Minor editing requred 

Author Response

Dear Reviewer, on behalf of all coauthors and myself, I would like to thank you for taking the time to read our manuscript and providing your valuable comments, allowing us to improve our manuscript's quality. Based on your comments we provided the following corrections: 

  1. please use the full form of PBF-LB/M AlSi7Mg0.6

    Ad.1. We changed the title as suggested. The present form is: "Microstructural investigation of process parameters dedicated to Laser Powder Bed Fusion of AlSi7Mg0.6 alloy "

  2. Abstract- The conclusion “The results of this study can be used to optimize the print-20 ing parameters for producing high-quality AlSi7Mg0.6 components using PBF-LB/M technology.” Needs to be revised. The conclusion should be the specific outcome of the study rather than a general recommendation.

    Ad.2. We rephrased this sentence by providing a specific outcome (minimized porosity level). "The statistical model created based on porosity investigation, allowed for illustration of the technological window and showed certain ranges of parameter values at which the porosity of the produced samples will be at a possible low level.

  3. Methodology- Add a description of “The 10 mm cube-shaped test bodies and printing conditions were created in Magic 19 software (Leuven, Belgium, Materialise).”

    Ad.3. We changed this sentence into two, more specific: The 10 mm cube samples were designed in Magic 19 software (Leuven, Belgium, Materialise), and the same software was used to provide printing parameters, such as laser power, layer thickness, exposure velocity, and hatching distance (clearance between laser irradiation lines). 

  4. Please revise this paragraph “For the experiment was used aluminum alloy AlSi7Mg0.6 (AC - 42200) was from SLM Solutions with a spherical particle size of 20 - 63 μm. The thermal conductivity of this material is 150 - 170 W/(m·K). The chemical compo- sition of this material according to the material data sheet is given in Table 1 below. The quality of the metal powder was checked by scanning electron microscope (SEM) Jeol JSM-6610 (Jeol, Tokyo, Japan).

    Ad.4. Thank you for pointing out this issue. We rephrased it: An aluminum alloy AlSi7Mg0.6 (AC - 42200), gas atomized powder, supplied by the SLM Solutions (SLM Solution, Lubeck, Germany) was used as a base material. The powder particles were characterized by a spherical particle size of 20 - 63 µm. The thermal conductivity of this material is 150 - 170 W/(m·K).

  5. The author has listed 37 printing parameters in Table 2. Does it mean the number of test groups is 37? How many specimens were tested in each group (n=?). The reviewer’s opinion the number of specimens in each test condition should be a minimum of 3 to enable statistical comparison and to draw a specific conclusion.  

    Ad.5. The porosity was calculated by using merged microscopic images from the possible highest area of the sample. Please see Figure 3, there are visible squares in the image - these are exact images taken by microscope during one measurement. So to combine this image for porosity measurement, about 56 images were taken into account for a single printing parameter. 

Reviewer 2 Report

Comments and Suggestions for Authors

This manuscript investigated the effect of process parameters on the microstructural properties of an AlSi7Mg0.6 alloy. The porosity and the hardness of the samples were tested and compared. It has practical benefits for additive manufacturing. It can be published after improvement. The suggestions are listed below: 

1. At the end of the introduction section, the tasks of this work should be claimed clearly. It is suggested to have a separate paragraph to describe the novelty and the tasks.

2. P4, L125. It says "the YZ plane...", but no coordinate system was provided.

3. P5, L161. What is y in Equation 2? After reading the whole manuscript, I guess it is porosity. However, it should be explained clearly when it first occurs.

4. P12, L299. "The dashed red line indicates the mean value for all measurements" It is confused here. All the measured hardness shown in Figure 14 are below the dashed red line, then, how can it be the mean value?

5. P13, Table 5. Why some porosity calculated by the statistic model equation (3) are missing?

Author Response

Dear Reviewer, on behalf of all coauthors and myself, I would like to thank you for taking the time to read our manuscript and providing your valuable comments, allowing us to improve our manuscript's quality. Based on your comments we provided the following corrections: 

  1. At the end of the introduction section, the tasks of this work should be claimed clearly. It is suggested to have a separate paragraph to describe the novelty and the tasks.

    Ad.1. Thank you for this comment. We made additional paragraphs and created a clear aim of our work: "The aim of our research is to develop a comprehensive understanding of the relationship between energy density and various printing parameters in the production of aluminum alloy components using PBF-LB/M technology. Specifically, we aim to investigate how different combinations of printing parameters affect the porosity and microstructure of AlSi7Mg0.6 alloy samples. By analyzing the influence of energy density along with other printing parameters, we seek to develop a mathematical model that accurately predicts the porosity levels and microstructural characteristics of printed components. This research is essential for optimizing the manufacturing process and achieving components with superior material and mechanical properties."

  2. P4, L125. It says "the YZ plane...", but no coordinate system was provided.

    Ad.2. Thank you for pointing out this issue. We rephrased this part: "For experiments and evaluation, samples were cut in the cross-section through the whole layers of each sample by means of a metallographic saw with direct cooling.

  3. P5, L161. What is y in Equation 2? After reading the whole manuscript, I guess it is porosity. However, it should be explained clearly when it first occurs.

    Ad.3. Of course, it is a porosity. We put proper explanation. 

  4. P12, L299. "The dashed red line indicates the mean value for all measurements" It is confused here. All the measured hardness shown in Figure 14 are below the dashed red line, then, how can it be the mean value?

    Ad.4. During uploading the manuscript the line moved above the chart. Now we fixed it to avoid such an issue. A proper version is in the corrected file. 
  5. P13, Table 5. Why some porosity calculated by the statistic model equation (3) are missing?

    Ad.5. The model required parameters that had a closed value of given variables. The use of the group of parameters that were not taken into account in the statistical model was only for validation - it was put into the manuscript. 

Round 2

Reviewer 1 Report

Comments and Suggestions for Authors

Thank you for the revision. No further comments. Good luck

Comments on the Quality of English Language

minor editing in the final version is required